# Improving the quality of neural TTS using long-form content and multi-speaker multi-style modeling

*Tuomo Raitio, Javier Latorre, Andrea Davis, Tuuli Morrill, Ladan Golipour*

Apple

## Abstract

Neural text-to-speech (TTS) can provide quality close to natural speech if an adequate amount of high-quality speech material is available for training. However, acquiring speech data for TTS training is costly and time-consuming, especially if the goal is to generate different speaking styles. In this work, we show that we can transfer speaking style across speakers and improve the quality of synthetic speech by training a multi-speaker multi-style (MSMS) model with long-form recordings, in addition to regular TTS recordings. In particular, we show that 1) multi-speaker modeling improves the overall TTS quality, 2) the proposed MSMS approach outperforms pre-training and fine-tuning approach when utilizing additional multi-speaker data, and 3) long-form speaking style is highly rated regardless of the target text domain.

**Index Terms**: Speaking style modeling, multi-speaker modeling, long-form data, neural TTS

## 1. Introduction

Neural text-to-speech (TTS) [1–3] can provide quality close to natural speech if an adequate amount of high-quality speech material is available for training. However, conventional TTS recordings usually consist of a single style, or the style variation is subtle. There is an increasing demand to generate speech with various speaking styles to provide better listening experience for various scenarios and content. The conventional approach of recording more speech data with desired speaking styles can provide a solution, yet such recordings are costly and time consuming, or sometimes infeasible if the speaker cannot perform a specific style or the speaker is not available for such recordings. A preferable and more scalable approach is to learn the styles from voices where such styles already exist, and then transfer the styles to target voices.

A speech utterance can be defined as the result of three main components: 1) the linguistic content, 2) speaker identity, and 3) prosody or speaking style. Here we define the speaking style as the remaining aspects after the linguistic content and the effect of speaker have been accounted for. The style covers various prosodic aspects such as speaking rate, pitch, loudness, and voice quality. In this work, we are interested in high-level speaking styles rather than individual prosodic features or their fine-grained structure. Speaking style implies how speech is expressed related to, for example, content and context, such as in long-form reading or conversational speaking style, emotional state of the speaker, such as happy or sad, or environment, such as in Lombard speech. The contribution of content, speaker, and speaking style to the speech signal are inherently entangled, and the main goal of style modeling is to disentangle these factors.

Various methods have been proposed to model speaking style in neural TTS. Some methods aim for fine-grained prosody transfer using a reference utterance with matching text [4, 5]. These methods are mostly applicable for offline tuning of individual sentences. Some methods learn a prosodic space using acoustic features that can be used to vary and change speaking styles [6–9], however, these methods are often inflexible to provide a general solution to style modeling. Many speaking style modeling methods learn a latent embedding space, derived from a reference acoustic representation containing the prosody [10–13]. This enables speaking style transfer by using a prosody embedding extracted from a reference utterance, or alternatively using self-supervised learning to cluster the embedding space into different styles. The downsides of these methods are that either they require a reference utterance (albeit matching text is not required), in which case there may be content or speaker leakage to the target utterance, or in the case of learned style embeddings, they may not correspond to desired styles by human listeners and may contain an undesired mix of styles and other acoustic factors. Overall, the disentanglement between content, speaker, and style is a difficult problem without large amounts of labeled speech data.

In this work, we investigate simple supervised speaking style modeling that relies on multi-speaker multi-style (MSMS) modeling using explicit speaker and style labels. The assumptions behind the proposed method are that 1) there exists speech recordings with at least two distinct speaking styles, 2) there are at least two speakers for each speaking style, 3) parallel speaking style data is not required for any speaker.

We investigate two distinct styles in this study: 1) a style aimed for general TTS and voice assistant purposes, and 2) long-form speaking style that is suitable for listening to audiobooks, webpages, and other long-form content. We utilize multiple speakers from each style and investigate if we can reproduce the styles with a speaker that has no recordings of that particular style. We perform extensive subjective experiments to measure the quality of the proposed MSMS method with different styles, and compare it to multi-speaker, pre-train fine-tune, and single-speaker based systems. We also measure the speaking style similarity between systems to assess the style transfer capability of the proposed method.

### 1.1. Relation to prior work

Speaking style modeling and transfer is a widely researched topic [10–27]. Our aims in this work are similar to the ones in [23–26] that use multi-speaker multi-style modeling, except that our approach uses fully supervised methods for disentangling the content, speaker, and style. Our work is also related to the studies using multi-speaker TTS that show quality improvements on multi-speaker datasets [5, 28–33]. In addition,

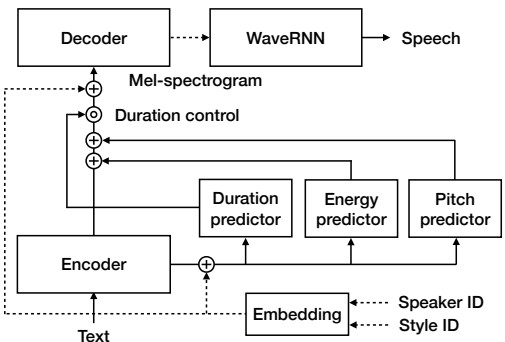

Figure 1: *Neural TTS architecture with the proposed speaker and style modeling (dashed line) over the baseline model (solid line).*

our work is related to the studies that aim for better long-form reading style [26, 27, 33].

Our contribution in this work is 3-fold. First, we show that we can improve neural TTS quality by using long-form content and multi-speaker multi-style modeling, and second, we show that multi-speaker multi-style modeling is the preferred way to leverage such data in Transformer-based TTS, over more conventional methods, such as pre-training and fine-tuning [34]. Although multi-speaker or multi-style modeling using supervised speaker and style labels are not novel as such, we demonstrate that we can achieve substantial quality gains using such methods. Third, we observe that long-form speaking style is highly rated even in non-long-form text domain, being similar or even preferred in comparison to a more traditional speaking style specifically aimed for TTS.

## 2. Multi-speaker multi-style modeling

We propose a simple but effective method for supervised simultaneous speaker and style modeling. The proposed method is based on combining knowledge from various speakers and styles without requiring parallel style data from any speaker. We condition the neural TTS model with 1-hot encodings of speaker and style, which enables switching the speaker and style at inference time. We call our method multi-speaker multi-style (MSMS) modeling.

### 2.1. Neural TTS architecture

Our neural TTS system, shown in Fig. 1, consists of a Transformer-based non-autoregressive acoustic model similar to FastSpeech 2 [3], and an autoregressive vocoder similar to WaveRNN [35]. The input to the acoustic model is a phoneme sequence with punctuation and word boundaries, and the output is a Mel-spectrogram. The model is based on a feed-forward Transformer (FFT) [3, 36] encoder and dilated convolution decoder. The encoder consists of an embedding layer that converts the phoneme sequence to phoneme embeddings followed by a series of FFT blocks that take in the phoneme embeddings with positional encodings and output the phoneme encodings. Each FFT block consists of a self-attention layer [36] and 1-D convolution layers along with layer normalization and dropout. The phoneme encodings are then fed to the variance adaptors that predict phone-wise duration, pitch, and energy. The variance adaptors consist of 1-D convolution layers along with layer normalization and dropout similar to [3]. Instead of using pitch spectrograms as in [3], we use continuous pitch, quantization,

Table 1: *Speech data.*

| Voice | Style | Dur. | Mdn. pitch | Ave. sent. len. |
|---|---|---|---|---|
| Voice 1 | TTS | 37 h | 188 Hz | 3.48 s |
| Voice 2 | TTS | 23 h | 112 Hz | 3.29 s |
| Voice 3 | TTS | 13 h | 148 Hz | 3.30 s |
| Voice 4 | TTS | 11 h | 119 Hz | 2.42 s |
| Voice 5 | TTS | 12 h | 145 Hz | 3.19 s |
| Voice 6 | Long-form | 40 h | 159 Hz | 2.81 s |
| Voice 7 | Long-form | 24 h | 84 Hz | 2.33 s |
| 7 voices | 2 styles | 160 h | 143 Hz | 2.94 s |

and finally projection to an embedding. The predicted phone-wise pitch and energy features are then added to the phoneme encodings, after which they are upsampled according to the predicted phone-wise durations. The decoder consists of a series of dilated convolution stacks instead of the original FFT blocks in [3], which improves model inference speed as well as saving runtime memory compared with the original design. Finally, the decoder converts the adapted encoder sequence into a Mel-spectrogram sequence in parallel.

To generate speech samples from the Mel-spectrogram, we use an autoregressive recurrent neural network (RNN) based vocoder, similar to WaveRNN [35]. The model consists of a single RNN layer with 512 hidden units, conditioned on Mel-spectrogram, followed by two fully-connected layers ($512 \times 256$, $256 \times 256$), with single soft-max sampling at the output. The model is trained with pre-emphasized speech sampled at 24 kHz and $\mu$-law quantized to 8 bits for efficiency. More information about the implementation can be found in [37].

### 2.2. Proposed system

On top of the baseline model, we add speaker and style conditioning for the variance adaptors and the decoder. We form a single combined embedding for the speaker and style as follows. We form 1-hot vectors of speaker and style (each size of 64), and then concatenate these together. We tile the embedding to each frame in time and feed through a dense layer to project to the corresponding output size, after which we add the resulting embedding to the input of the variance adaptors and the decoder. At inference time, we can generate speech with any speaker and any style, regardless of whether the combination was seen in the training data.

## 3. Experiments

### 3.1. Data

We use proprietary speech data from a total of seven American English speakers to train our models. The speech data for five speakers (voices 1–5) is recorded aimed at voice assistant purposes, consisting of assistant dialog, navigation, but also some material from books and Wikipedia, for example. We refer this style as TTS style. The speech data from the remaining two speakers (voices 6–7) are long-form recordings of various fictional books. The most notable difference between the two styles investigated here is that the TTS style is recorded solely for TTS development purposes, emphasizing clarity and pronunciation, while the long-form style is recorded for audiobook purposes, emphasizing narration. More details about the speech data are presented in Table 1, including total duration of each dataset, median pitch of the voices, and average sentence lengths. For testing our models, we synthesize speech using 300 sentences, consisting of 75 sentences of 4 types: 1) books, 2) knowledge, 3) navigation, and 4) dialog. Specifically, the books

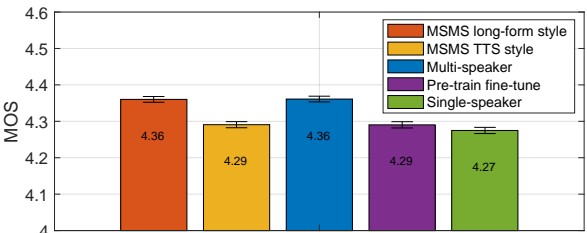

Figure 2: *Overall MOS results with 95% confidence intervals.*

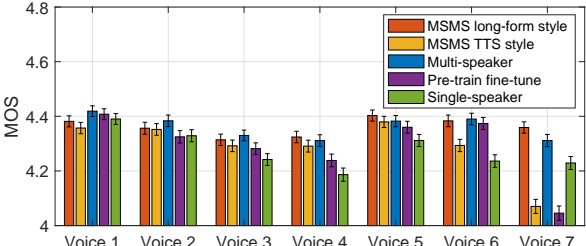

Figure 3: *MOS results per voice and system with 95% confidence intervals.*

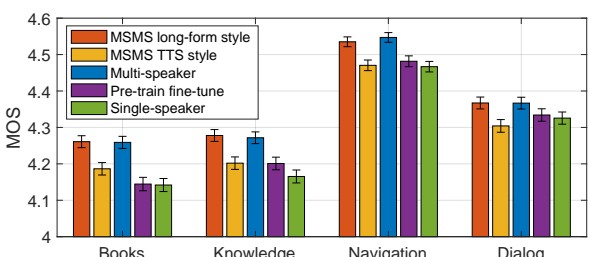

Figure 4: *MOS results per sentence type and system with 95% confidence intervals.*

category consists of long-form fiction book content, knowledge category consists of short answers to factual questions, navigation category consists of navigation guidance sentences, and the dialog category consists of digital assistant dialog sentences.

### 3.2. Model training

We use the following systems in our evaluation:

1. **MSMS long-form style:** Proposed system conditioned on speaker and style, inference with long-form speaking style.

2. **MSMS TTS style:** Proposed system conditioned on speaker and style, inference with TTS style.

3. **Multi-speaker:** Proposed system conditioned only on speaker, style conditioning kept constant.

4. **Pre-train fine-tune:** Pre-training with all data and fine-tuning with target speaker, no speaker or style conditioning.

5. **Single-speaker:** Training with target speaker data, no speaker or style conditioning.

We trained all the models for 140k steps using 16 GPUs and a batch size of 512, except for the pre-train fine-tune model where we first pre-trained the model for 200k steps with 16 GPUs and then fine-tuned the model for 10k steps using a single GPU and target speaker's data. No model architecture or size changes were made between the systems 1–5 other than the ones mentioned in Sec. 2.2 to add speaker and style conditioning. The WaveRNN models [37] to generate speech from the Mel-spectrograms were trained separately for each speaker.

We use phone-wise duration, pitch, and energy as the fine-grained features. 80-dimensional Mel-spectrograms are computed from pre-emphasized speech using STFT with 25 ms frame length and 10 ms shift. The encoder has 4 feed-forward Transformer layers each with a self-attention layer having 2 attention heads and 256 hidden units, and two 1-D convolution layers each having a kernel size of 9 and 1024 filters. The decoder has 2 dilated convolution blocks with six 1-D convolution layers with dilation rates of 1, 2, 4, 8, 16, and 32, respectively, kernel size of 3, and 256 filters. The feature predictors have two 1-D convolution layers with kernel size of 3 and 256 filters. We use dropout rate of 0.2 and layer normalization with $\epsilon = 10^{-6}$.

### 3.3. Quality

We evaluated naturalness using a 5-point mean opinion score (MOS) test. We conducted 7 separate listening tests, one for each voice, to keep the listening task manageable. We synthesized 300 utterances for each system and voice, consisting of 75 sentences of 4 types (books, knowledge, navigation, dialog). After the synthesis, the average durations of the utterances were 4.67, 8.03, 3.49, and 3.06 seconds for the books, knowledge, navigation, dialog categories, respectively. The average dura-

tion of all evaluation utterances was 4.81 seconds. Overall, 651 American English native speakers participated in the tests using headphones (54.4 %) or loudspeakers (45.6 %), and gave a total of 157,500 ratings, consisting of 7 voices × 5 systems × 300 utterances × 15 ratings.

The overall MOS results per system are depicted in Fig. 2, showing that MSMS long-form style and multi-speaker systems have the highest overall ratings. The results per voice and system are shown in Fig. 3, showing that there are voice-dependent differences in the results, especially for voice 7 where MSMS TTS style and pre-train fine-tune systems were rated lower. The results per sentence type are shown in Fig. 4, showing that navigation domain is rated the highest overall, but still showing the general trend in the overall performance of the systems, regardless of the sentence type. The score distributions per system are shown in Fig. 5, showing a higher proportion of 5's for the MSMS long-form style and multi-speaker systems. The detailed results are shown in Fig. 6.

Linear mixed models with fixed effects of sentence type, voice, and speaking style and random effects of listener, item, and listening device were constructed. A likelihood ratio test showed that the model with a three-way interaction between speaking style, sentence type, and voice was a significantly better fit than a model with two-way interactions of the fixed ef-

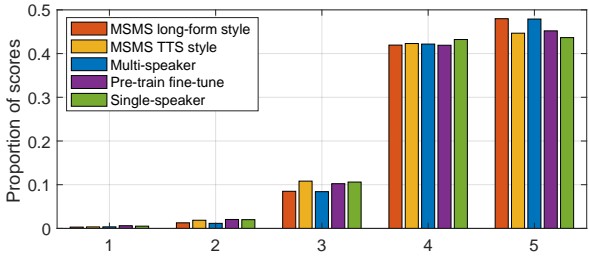

Figure 5: *MOS distribution per system.*

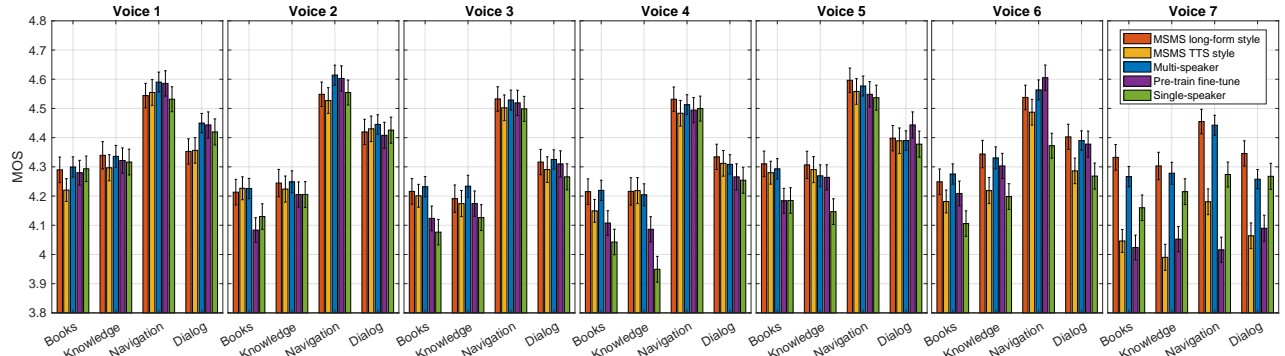

Figure 6: *Detailed results of the subjective evaluation: speech quality MOS per voice, target text domain, and system with 95% confidence intervals.*

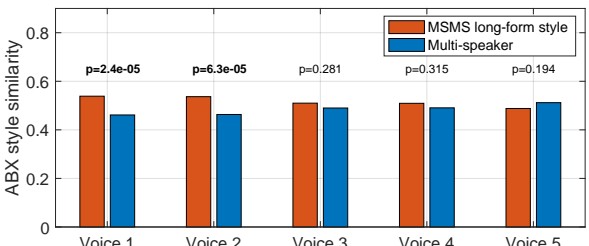

Figure 7: *Results of the ABX speaking style similarity test between MSMS long-form style and Multi-speaker systems, where the reference style is natural long-form samples. Statistically significant results ($p \leq 0.05$) are marked with bold p-values.*

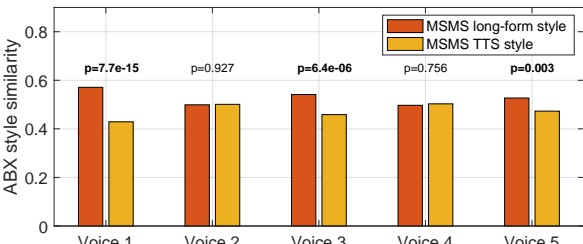

Figure 8: *Results of the ABX speaking style similarity test between MSMS long-form style and MSMS TTS style systems, where the reference style is natural long-form samples. Statistically significant results ($p \leq 0.05$) are marked with bold p-values.*

fects ($\chi^2$ = 153.76, p < 0.0001). The effect of speaking style on listener rating depended on the sentence type and the voice. Pairwise comparisons showed that a) multi-speaker was always rated higher than single-speaker across voice and sentence type, in nearly all cases significantly, b) multi-speaker was always rated higher than pre-train fine-tune across voice and sentence type, in most cases significantly, c) MSMS long-form was rated higher than pre-train fine-tune in most cases across voice and sentence type, in most cases significantly, d) MSMS long-form style was rated significantly higher than MSMS TTS style for some but not all voices across sentence type, with no case of significant preference for MSMS TTS style in any sentence type, e) pre-train fine-tune was largely not significantly different from single-speaker, with the exception of voice 7 where single-speaker was rated significantly higher than pre-train fine-tune.

### 3.4. Speaker and style similarity

The aim of the proposed approach is to retain speaker similarity of the original speaker while reproducing the desired styles. In order to assess speaker similarity between the systems, we calculated 256-dimensional speaker embeddings [38] of each of the 300 synthesized utterances for each system and speaker, and then calculated the mean squared error (MSE) and cosine similarity of the sentences between the systems. The single-speaker system is used as the reference as it uses data only from the target speaker. The results, depicted in Table 2, show that the MSMS long-form style is closest to the single-speaker system.

Assessing the style of speech can be challenging. Using informal listening, the long-form speaking style was confirmed

to be reproduced and transferred to the target voices 1–5 in the MSMS long-form style system (to a varying degree), while the speaking style of the Multi-speaker system was perceived to be closer to the original speaking style of the voices, as is expected without style conditioning. In order to further investigate the speaking style, we performed ABX speaking style similarity tests, where listeners heard three samples, A, B, and X, and their task was to choose the sample, A or B, that is more similar in speaking style to the reference sample X.

We conducted two ABX comparisons: 1) MSMS long-form style vs. Multi-speaker, and 2) MSMS long-form style vs. MSMS TTS style. The aim of the first speaking style similarity ABX test was to find out if the speaking style of the MSMS long-form style, that aimed to learn the style from the long-form material, is more similar to the long-form speaking style than with the Multi-speaker system that has not specifically learned the long-form style. The aim of the second ABX test was to

Table 2: *Speaker similarity of the systems in comparison to the single speaker system measured by MSE and cosine similarity of the speaker embeddings.*

| System | MSE ↓ | Cosine sim. ↑ |
|---|---|---|
| Single-speaker (reference) | 0.00 | 1.000 |
| MSMS long-form style | $8.20 \cdot 10^{-3}$ | 0.961 |
| Fine-tune pre-train | $9.73 \cdot 10^{-3}$ | 0.947 |
| Multi-speaker | $9.84 \cdot 10^{-3}$ | 0.946 |
| MSMS TTS style | $9.91 \cdot 10^{-3}$ | 0.946 |

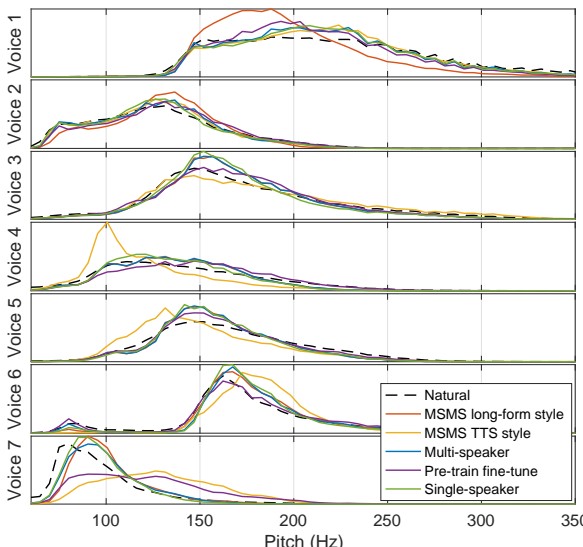

Figure 9: *Pitch distribution for each voice and system in comparison to natural speech.*

confirm that the MSMS long-form style system reflects the intended long-form style more than the MSMS TTS style system.

We used 3,000 sample pairs (A/B) for each of the voices 1–5. As the reference samples X, we randomly assigned a natural recording of a long-form style sample from voices 6–7. Overall, 53 and 58 American English native speakers participated in the two tests, respectively, and gave a total of 30,000 ratings, consisting of 5 voices × 3,000 utterances × 2 ABX tests. The results of the first speaking style similarity ABX test are depicted in Fig. 7. Two-sided binomial test shows that the speaking style of the MSMS long-form style system is assessed to be more similar to the natural long-form samples than the Multi-speaker system for voices 1 and 2 (p ≤ 0.05), while for other voices there are no statistically significant differences. The results of the second ABX test are depicted in Fig. 8, showing that the speaking style of the MSMS long-form style system is assessed to be more similar to the natural long-form samples than the MSMS TTS style system for voices 1, 3, and 5 (p ≤ 0.05), while for other voices there are no statistically significant differences.

Assessing the similarity of speaking style or prosody is a difficult task, and there is no agreed standard for assessment. One example can be found in [39] where the speaking style was assessed within the same speaker. In the current study, the speaking style similarity had to be assessed across speakers, which made the task even harder. Despite the difficulty of evaluating speaking style similarity across voices, the ABX test results indicate successful speaking style transfer for some voices, and especially for the voice 1 that was originally very dissimilar to the long-form reading style.

To further investigate the effects of style, we calculated the distributions of pitch for each voice and system. The results, depicted in Fig. 9, show that the pitch distributions mostly follow the original data, while there are also some interesting differences. For example, MSMS long-form style for voice 1 shows a lower pitch distribution, which is also perceived as calmer and closer to the long-form style in contrast to the original TTS style. Several voices also show large differences in pitch distribution for the MSMS TTS style, such as voices 4–7, which are

perceived less similar to the original speaking style. The reason for this may be due to the varied style data of the TTS style recordings (see Sec. 3.1), which makes it harder for the model to learn a consistent style.

## 4. Discussion

The results indicate that including long-form content and using multi-speaker modeling, either with or without style modeling, improves the overall quality. This is shown by the improved MOS ratings for the MSMS long-form style and multi-speaker systems that were rated better than the single-speaker system. The finding is not surprising as adding more data with multi-speaker modeling has been shown to improve TTS quality [30, 31, 33]. The results also indicate that MSMS modeling can be a better way to utilize additional multi-speaker speech content in improving TTS quality over the pre-train and fine-tune method [34]. This makes sense as pre-training and fine-tuning has a tendency for catastrophic forgetting [40], while MSMS modeling does not have such a problem. The study also shows that long-form style is highly rated regardless of the text domain. The long-form style seems to be rated equal or better in comparison to the traditional TTS style, even in dialog and navigation domains although the TTS style is specifically targeted for those domains. This is a somewhat surprising finding that may have implications for future TTS recordings.

In future work, we aim to expand the investigation by adding more speakers and styles to confirm the generalizability of the approach. Finally, the surprising results regarding high quality of the long-form style for general TTS, regardless of the domain, deserves further investigation. The current subjective evaluations were performed without the real-life context for the sentences, and we aim to investigate if the positive results still exist in actual applications.

## 5. Conclusions

We proposed a simple supervised multi-speaker multi-style (MSMS) TTS modeling framework that enables speaking style transfer across speakers, and demonstrated that we can improve the quality of existing TTS voices by using long-form recordings from new speakers. In particular, we showed in extensive evaluations that 1) multi-speaker modeling improves the overall TTS quality, 2) the proposed MSMS approach outperforms pre-training and fine-tuning approach when utilizing additional multi-speaker data, and 3) long-form speaking style is highly rated regardless of the target text domain.

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
