# OpenReview forum: "Improving the quality of neural TTS using long-form content and multi-speaker multi-style modeling"
_Interspeech.org/2023/Workshop/SSW — SSW12_

### Official Review · Reviewer_V182 · 2023-05-17
**An analysis of an in-house TTS system's training procedures across several speaker datasets for long form synthesis style transfer**

**Rating:** 6
**Confidence:** 5

**Review:**

Strengths:
The paper introduces a set of analyses over a multi speaker multi style FastSpeech2-like architecture's performance for long-form style transfer. The analyses are performed with several large listening tests containing 7 voices and two speaking styles (TTS-like and long form) across 5 training procedures.  The results are extensive and show variances across the speakers and training procedures' ratings. The TTS system and training data are in-house.

Weaknesses:
- it is not clear if changes were made to the TTS architecture to accommodate the diferent training set sizes, especially for the large amount of data present in the MSMS system. The FastSpeech architecture is not known to discriminate speaker/style identities very well.
- I cannot find any natural samples' ratings in the listening test
- I do not understand the choice of 64 dim one-hot embedding if there are only 7 voices and 2 styles
- More details should be included wrt the gender, accents and timbre similarities of the speakers.
- there are too many Figures representing more or less the same information. Using a boxplot (instead of bar plot) in Figure 6 and perhaps having two smaller figures would have helped with the clarity of the results.
- what is the average utterance duration in each scenario (training and evaluation-wise)?

Questions/potential improvements:
- The speaker similarity table could have included the similarity wrt natural samples
- Is there a correlation between the high ratings of navigation samples and the familiarity of the listeners with the TTS being used in this scenario for a very long time now. As opposed to the other domains...
- I would have liked to see an analysis of the correlation between the amount of training data for each speaker and its ratings
- It seems rather strange to me that style transfer works better than the default style of the speaker. Any additional insights into this?
- Speaker 7 has the lowest pitch which can cause problems in current TTS and vocoder architectures. You may want to look into this, as well.
- is the identity of the speaker consistent across the synthesised utterances in style transfer scenarios? or can it change identity from one sample to the other?

Editing suggestions:
- compress the list of references in section 1.1 to intervals (e.g. [20-27]).
- cite the papers in the order of their appearance (i.e. [11,22,33], not [22,33,11]
- please go through the paper again, there are some minor typos

---

> ### Author Response · Authors · 2023-06-23
> **Thank you for the valuable feedback and comments**
>
> Re. "it is not clear if changes were made to the TTS architecture to accommodate the different training set sizes, especially for the large amount of data present in the MSMS system." No changes were made between the systems 1-5 except the ones mentioned in Section 2.2 to add speaker and style conditioning. We have clarified this in the final version of the paper.
>
> Re. "I cannot find any natural samples' ratings in the listening test". It was a deliberate decision to limit the number of systems in the test and not include natural samples in the test. The rationale for this is that the focus of the study was to compare the systems in the test to each other, and including natural samples would have compressed the MOS ratings and therefore would have made the comparison more difficult.
>
> Re. "I do not understand the choice of 64 dim one-hot embedding if there are only 7 voices and 2 styles". There is no specific meaning for this dimension except for allowing more speakers in other data settings, and in this case the dimension could be smaller.
>
> Re. "More details should be included wrt the gender, accents and timbre similarities of the speakers". We have provided the median pitch of the speaker in Table 1, from which the reader can infer some of the speaker characteristics. The authors intentionally do not mention the genders of the voices in this work, but the vocal range can be inferred from Figure 9. All voices were from American English speakers.
>
> Re. "there are too many Figures representing more or less the same information. Using a boxplot (instead of bar plot) in Figure 6 and perhaps having two smaller figures would have helped with the clarity of the results." Figures 2-6 all represent different angle on the results, and their purpose is to provide any necessary information to the reader. The authors agree that there is a lot of information in Figure 6, however, it should be rather easy to zoom and read the figure on electronic devices.
>
> Re. "what is the average utterance duration in each scenario (training and evaluation-wise)". The average durations of the training utterances are as follows: 3.48, 3.29, 3.30, 2.42, and 3.19 seconds for the TTS style voices 1-5, and 2.81 and 2.33 seconds for the long-form style voices 6 and 7. Overall, the average training utterance duration was 2.94 seconds. The average durations of the evaluation utterances are as follows: 4.67 seconds for books, 8.03 seconds for knowledge, 3.49 seconds for navigation, and 3.06 seconds for dialog. Overall, the average evaluation utterance duration was 4.81 seconds. We have added this data to the final version of the paper.
>
> Re. "The speaker similarity table could have included the similarity wrt natural samples". In order to get accurate results, it was important to evaluate speaker similarity using matching sentences. We did not have the ground truth recorded sentences for the evaluation utterances, and therefor using natural samples was not feasible.
>
> Re. "Is there a correlation between the high ratings of navigation samples and the familiarity of the listeners with the TTS being used in this scenario for a very long time now. As opposed to the other domains...". This may be true, but it may also be so that these are prosodically rather simple sentences, and therefore it is easy for the TTS system to get them correct.
>
> Re. "I would have liked to see an analysis of the correlation between the amount of training data for each speaker and its ratings". The authors agree that this would have been useful and interesting, but this is left as future work.
>
> Re. "It seems rather strange to me that style transfer works better than the default style of the speaker. Any additional insights into this?". This may have been one of the most important findings of the study. As stated in the paper: "long-form speaking style is highly rated regardless of the target text domain", it seems that the listeners may prefer long-form style for general TTS output rather than the more traditional and controlled declarative style.
>
> Re. "Speaker 7 has the lowest pitch which can cause problems in current TTS and vocoder architectures. You may want to look into this, as well." Based on Figure 9, pre-train fine-tune and MSMS TTS Style system did have some trouble with this speaker since the pitch distributions are not that close to the original ones.
>
> Re. "is the identity of the speaker consistent across the synthesised utterances in style transfer scenarios? or can it change identity from one sample to the other?". The speaker identity is consistent in the style transfer scenarios except for the cases where pitch was not accurately transferred (see above comment), but even in those cases the vocal tract was correctly modeled.
>
> Re. "compress the list of references in section 1.1 to intervals (e.g. [20-27])" and "cite the papers in the order of their appearance (i.e. [11,22,33], not [22,33,11]". We have addressed these comments in the final version of the paper.

---

### Official Review · Reviewer_xtqJ · 2023-06-08
**Review -  style modeling in neural TTS**

**Rating:** 5
**Confidence:** 4

**Review:**

The proposed study addresses the problem of style modeling in neural TTS and speaking style transfer across speakers. The proposed system is based on the supervised MSMS technique (multi-speaker multi-style). A one-hot encoding vector encoding style and speaker is used to bias/condition the variance adaptor and the decoder of a Fastspeech2-like architecture.

Key Strength of the paper

Since the proposed MSMS is not novel (as mentioned by the authors), the main contribution of the paper is its extensive evaluation using MOS and ABX listening tests. This evaluation reveals that the MSMS approach is particularly suitable for processing long-form speaking style.

Main Weakness of the paper

The paper does not describe with enough details the specificities at the prosodic level of each of the 4 speaking styles considered (books, knowledge, navigation, and dialog). For instance, what is the expected difference between « knowledge » and « books »?  Are there really different prosodic patterns between the different styles ? Similarly, the concepts of « long-form style » and « TTS style » are not well defined. Why Wikipedia is considered as  long-form  (the the best of my understanding) and not « dialogue »? Does long-form means « beyond the current sentence »? In addition, as the proposed method is fully supervised, it might have been interesting to assess the degree of consensus on style labeling (e.g. by reporting a measure of inter-rater agreement on the training data). This lack of information about the content of the data set and the prosodic specificities expected of each style prevents the reader from assessing the real difficulty of the task (and therefore the improvements brought about by the MSMS technique).

Novelty/Originality :

As mentioned by the authors, MSMS is not novel. However, an extensive evaluation of this technique is of interest for the SSW audience.

Technical Correctness :

The subjective tests along with their  statistical analyses are well conducted. However, with a proprietary dataset and a lack of detail about its content, this research is difficult to be reproduced and the scientific impact of this paper is therefore limited.

Suggestions for improvement

  Providing a better definition of TTS vs. long-form styles together with a more fine-grained analysis of the prosodic content of the training data for each style considered.

Quality of References  : Good enough

Clarity of Presentation : The  paper is well written.

---

> ### Author Response · Authors · 2023-06-23
> **Thank you for the valuable feedback and comments**
>
> Regarding the comment about "the prosodic level of each of the 4 speaking styles considered (books, knowledge, navigation, and dialog)", these are not actually speaking styles but only a classification of input text into these categories. To be more specific, the books category consists of long-form fiction book content, the knowledge category consists of short answers to factual questions, the navigation category consists of navigation guidance sentences, and the dialog category consists of digital assistant dialog (short) sentences. We have added more information to the final version of the paper. Regarding long-form style vs. TTS style in the actual training data, the most significant difference is that the TTS style is recorded solely for TTS development purposes while the long-form style is recorded for audiobook purposes. We have added more information to the final version of the paper.

---

### Decision · Program_Chairs · 2023-06-14

**Decision:**

Accept

**Comment:**

SSW2003 received 45 papers. The acceptance rate is 82%. We are pleased to inform you that your paper has been accepted by the SSW2023 Program Committee. Please read the reviews carefully and submit your camera-ready paper by June 28th. Most of reviewers performed a detailed review. Please answer to their questions and take into account their comments.
Since your paper received a score below 5/9 that is strongly argued by the reviewers, note that the Program Committee will check if your manuscript has been significantly changed to specifically consider their remarks. Note that camera-ready papers are credited with one extra page to allow authors to consider reviewers’ suggestions. So max 7 pages in total including figures & refs.
The deadline for submitting the revised version (with full non anonymized authors and refs!) is 28th June.